# Urinary Tract Infection in Febrile Children with Sickle Cell Disease Who Present to the Emergency Room with Fever

**DOI:** 10.3390/jcm9051531

**Published:** 2020-05-19

**Authors:** Nehal Patel, Ahmad Farooqi, Michael Callaghan, Usha Sethuraman

**Affiliations:** 1Division of Emergency Medicine, Carman and Ann Adams Department of Pediatrics, Children’s Hospital of MI, Detroit, MI 48201, USA; usethura@dmc.org; 2Children’s Research Center of MI, Wayne State University School of Medicine, Detroit, MI 48201, USA; afarooq@med.wayne.com; 3Division of Pediatric Hematology and Oncology, Carman and Ann Adams Department of Pediatrics, Children’s Hospital of MI, Detroit, MI 48201, USA; mcallagh@med.wayne.edu

**Keywords:** sickle cell disease, urinary tract infection, fever, children

## Abstract

Sickle cell disease (SCD) patients are thought to be at higher risk for urinary tract infections (UTIs) compared to the general population secondary to increased sickling, abnormal urinary acidification, and an inability to concentrate the urine. The incidence of UTI in febrile children with SCD in the United States is unknown. Our objectives were to determine the rate of UTI among febrile SCD children and describe the risk factors for UTI in this population. We conducted a retrospective chart review of all febrile SCD patients <4 years of age who presented to a pediatric emergency department from 2012–2017 and who had a sterile sample of urine for analysis. A total of 167 febrile patients with SCD with 464 visits were identified. The majority were African American (95.2%), female (58.7%), and had hemoglobin SS (HbSS) (65.3%). The rate of UTI was 4.1%. All patients with a UTI were African American females with a median age of 19 months (IQR 12–30). On regression analysis, no risk factors were associated with a UTI. We found the rate of UTI in febrile young children with SCD was comparable to non-SCD children. Larger studies are required to identify the presence of risk factors for UTI in this population.

## 1. Introduction

Sickle cell disease (SCD) is one of the most common genetic hematological diseases, with an estimated 100,000 Americans affected [1]. Disease severity varies considerably, and important complications include invasive infections and sepsis. Due to the subnormal immunity resulting from reduced or absent splenic function, children with SCD are at a higher risk for bacterial infections [2,3]. However, patients with SCD also have altered blood flow to the kidneys and low renal arterial oxygen tension. This causes increased sickling, abnormal urinary acidification, and an inability to concentrate the urine, resulting in an increased risk for urinary tract infections (UTIs) [3,4,5]. Yet, studies exploring the pediatric rates of UTI in SCD with fever are limited, and results have ranged widely between 1–28% [4,6,7,8,9,10].

The current practice in many pediatric emergency departments (PEDs) is to perform a urinalysis (UA) and a urine culture in SCD patients with fever as part of the diagnostic evaluation [11]. In children less than 2 years of age who are not able to provide a clean catch sample, a catheterized specimen is often required, as bag samples have a high rate of contamination [12]. With multiple PED visits for a chief complaint of fever, patients with SCD may undergo multiple urinary catheterizations, which can add to parental concerns for this invasive procedure. Knowledge of the actual rate of UTI in the SCD population and the predictors of a UTI in these patients would help to limit the diagnostic evaluation to those who are at a higher risk for a UTI. Thus, the aim of this study was to (1) determine the rate of UTI among children with SCD and fever and (2) describe the risk factors for UTI in this patient population.

## 2. Experimental Section

### 2.1. Study Setting

This study was performed at an inner-city tertiary care children’s hospital that serves as a referral center for patients with SCD. Approximately 700 patients with SCD are cared for in the pediatric hematology clinic and 150 are seen in the PED annually for fever. The PED is a level one trauma center with approximately 85,000 annual patient visits and is staffed by pediatric emergency medicine physicians, general pediatricians, and advanced practice providers. This study was approved by the Wayne State University Institutional Review Board (IRB Number: 123417MP2E) and the study was conducted in compliance with all applicable institutional ethical guidelines for retrospect research studies.

### 2.2. Study Sample

This was a retrospective descriptive study of children with SCD less than 4 years of age who presented to the PED with fever between 1 January 2012 and 31 December 2017. We chose this age group as they are less likely to provide a clean catch urine sample and would require a catherization. Patients were identified using International Classification of Diseases (ICD), Ninth and Tenth Edition diagnosis codes for sickle cell (282.60, 282.61, 282.62, 282.63, 282.64, 282.68, D57.0, D57.02, D57.1, D57.2, D57.219) and fever (780.60, 780.61, D50.9, R50.81). We excluded children who were >4 years of age and those who (a) did not have a chief complaint of fever; (b) received antibiotics other than for prophylaxis within 2 weeks prior to presenting to the PED; (c) had preexisting renal disease, a neural tube defect, or required home urinary catheterization; and (d) did not provide a sterile sample for urinalysis or urine culture.

### 2.3. Definitions

SCD was defined as hemoglobin SS, SC, S-beta thalassemia (S-beta thal), or SE as diagnosed by the newborn screen or hemoglobin electrophoresis. Fever was defined as tactile fever at home or any temperature ≥38 °C at home or in the PED. We defined the height of fever as the maximum temperature that the patient had during their febrile illness. Length of stay was defined as time measured from patient registration in triage to when they were discharged either from the PED or the hospital following admission. A sterile sample of urine was defined as a midstream clean catch, catheterized, or suprapubic tap sample.

The American Academy of Pediatrics (AAP) defines a UTI based upon method of collection: greater than 50,000 colony forming units per milliliter (CFU/mL) of an uropathogen for a sample obtained via catheterization or suprapubic aspiration with a positive UA (presence of pyuria and/or bacteria) [13]. Our laboratory reports urine CFU as less than 10,000, 10,000–100,000, or greater than 100,000 CFU/mL per pathogen. Due to this limitation, a UTI was therefore defined as a urine culture proven greater than or equal to 10,000 CFU/mL of at least one uropathogen.

### 2.4. Outcomes

The primary outcome for this study was the rate of UTI in children less than 4 years of age with SCD who presented to the PED with fever. The secondary outcome was risk factors associated with a UTI in our cohort of patients.

### 2.5. Data Collection

All data were collected and recorded into Research Electronic Data Capture (REDCap), a secure, web-based, application designed to support data capture for research studies [14]. Variables of interest included age, gender, race, type of SCD, height of fever, complete blood count with differential results, urinalysis results, urine culture results, and length of stay. All variables were abstracted by a trained researcher assistant and verified by the primary investigator.

### 2.6. Data Analysis

The data was summarized and reported with categorical variables by numbers and percentages. The rate was measured by the frequency of which a UTI occurred in a defined population over 6 years. This was calculated with number of UTIs over the total number of visits. The normality of continuous variables was tested by the Shapiro–Wilk test. We analyzed all normally distributed continuous variables by mean and standard deviation, whereas the non-normally distributed continuous variables were reported by median and interquartile range. Pearson’s Chi-squared test was used to analyze the distribution of categorical variables by groups, provided no expected frequency was less than 1, and no more than 20% of the cell had a frequency of less than 5. Otherwise, Fisher’s exact test was used for the analysis. Missing data accounted for less than 1% of the total data set. A total of five cases had at least one missing data point. Missing data were handled using pairwise deletion. Two group comparisons on UTI for non-normally distributed continuous variables such as age in months, maximum temperature (°C), and white blood cell count (WBC) were compared using the Wilcoxon rank sum test. Generalized estimating equation for logistic regression was used to study the effect of different predictors (age of months, WBC, and maximum temperature) on the likelihood of UTI for the same subjects with repeated visits. We used SAS (version 9.4, SAS Institute Inc. Cary, NC, USA) to perform the statistical analyses. The significance level was set at 0.05.

## 3. Results

In our study period, 212 patients with SCD had 897 PED visits for fever. Of these, 167 met our inclusion criteria (Figure 1). The characteristics of our study population are shown in Table 1. Over half of patients were female (58.7%), and the majority were African American (95.2%) with HbSS disease (65.3%).

There were 19 visits with 18 patients who met our definition for a UTI. The rate of UTI in SCD patients in our population was 4.1%. All of the patients with a UTI were African American females with a median age of 19 months and were treated with antibiotics during the visit. There were no differences in the median age of patients (19 months vs. 17 months, *p* < 0.42), white blood cell count (13.8k/cumm vs. 13.6k/cumm, *p* < 0.82), and maximum temperature (39.5 °C vs. 39.3 °C; *p* ≤ 0.13) between those with and without a UTI (Table 2). The urinalysis for patients with a UTI was more likely to be positive for leukocytes, nitrates, or an elevated urine white cell count (*p* < 0.001). There were three patients who had proteinuria seen in their urinalysis. The most common bacteria found in the urine was *Escherichia coli* (Table 3). Based on the generalized estimating equation for logistic regression analysis, age, maximum temperature, and white blood cell count were not associated with a UTI (Table 4).

## 4. Discussion

Our study showed that the rate of UTI in children <4 years of age with SCD was low and comparable to that reported among the general pediatric population [13,15]. It has been reported that children with SCD are at greater risk for UTI compared with the general population [4]. This increased susceptibility is felt to be due to altered blood flow, causing papillary necrosis and loss of the urinary concentrating and acidifying ability of the nephrons. Repeated red blood cell sickling and congestion in the vasa recta leads to ischemia and impaired solute reabsorption, resulting in poor concentrating ability. This results in abnormally dilute and alkaline urine and dehydration, which favor bacterial proliferation [16]. Despite these physiological differences in the renal function of patients with SCD, our study did not find that the rates of UTI in febrile children with SCD were any higher than that of the general population among children <4 years of age. While the reasons for this are not clear, one possible explanation may be that the decreased urinary concentration ability and hyperperfusion of the kidney that occur in young children with SCD may offer some amount of protection from urinary infections in this age group, as it can lead to increased urine output and voiding, which can result in less retention of bacteria and colonization or infection. While it would be interesting to see the effect of renal dysfunction on those with UTI in children with sickle cell disease, this was not within the scope of this emergency department based study.

Our rates of UTI are higher than that reported by other studies [7,17]. Differences in population, type of sickle cell disease, and age might have contributed to the reported difference in rate of UTI. Bansi et al. found an incidence of 1.1% for UTI; there were two patients that had a UTI—both were females and less than 24 months of age. However, their study cohort had included all children <18 years, and they did not collect a urine sample for each patient, which may have underestimated the incidence of UTI. Morrissey et al., in the United Kingdom, found a slightly higher incidence of UTI at 2.3%. However, not all patients in their study had a urine sample obtained, and their study included children <16 years of age; thus the true incidence of UTI may have been underestimated.

Conversely, our reported rate of UTI was much lower than that reported in tropical countries among patients with SCD [6,8,9]. Asinobi et al. described the prevalence in Nigeria to be 21.6%, with 62.2% having no symptoms or signs of UTI except for a fever [6]. Jiya et al. and Mava et al. also found similar results with incidences of 27.8% and 26%, respectively [8,9]. One of the proposed explanations for this observed difference in UTI rates is that in hotter areas, such as Nigeria, the decreased urinary frequency secondary to increased insensible loses promotes urinary retention and hence more time for bacterial pathogens to grow [8]. Although higher rates of UTI have been reported in adults in tropical countries due to differences in hygiene practice, the impact of cultural and population differences in hygiene practice on rates of UTI in children with sickle cell disease is unknown and needs to be explored further [18].

Young age and female gender are known risk factors for UTI in the general population [13,15,19]. Other studies on children with SCD have also reported gender differences in the rates of UTI in SCD [6,8,9]. While Asinobi et al. and Mava et al. reported a higher rate of culture-proven UTI in females (57%), Jiya et al. found higher rates in males (57%) [8]. Although we did not find any risk factors to be associated with a UTI in our study, it is noteworthy that the median age in our study among those with a UTI was 19 months and all were females. Larger studies are required to confirm age or gender differences of UTI among children with SCD and fever. If age and gender differences are found to be similar to that of the general population, then urine testing may be limited to those specific groups in SCD children with fever.

If our study results are confirmed in a larger sample, then clinicians could potentially obtain catheterized urine cultures only in the at-risk population of children with SCD using the same criteria as for the children without sickle cell disease. Additionally, a screening urine analysis in triage with a bagged urine specimen in the at-risk children could guide the need for an invasive catheterized urine culture [20].

There were several limitations to our study. This was a single center study with a small sample size, thus limiting our ability to make firm recommendations about evaluating patients with SCD for a UTI in clinical practice. However, our hospital is the largest referral center for SCD in the state and thus future multicenter studies are required to further increase the sample size. This was a retrospective study with the inherent limitations of chart review. Potentially eligible patients may have been excluded due to lack of documentation of temperature or method of urine collection. Thus, the true rate of UTI in febrile children less than 4 years of age could have been underestimated. Finally, our definition of UTI was limited by current laboratory reporting practices, which may, in turn, overestimate the rate of UTI, given the broad CFU counts used to define a UTI.

## 5. Conclusions

Despite the known risk for developing infections, the rate of UTI in febrile young children less than 4 years of age with SCD was comparable to non SCD children with fever. Larger studies are required to identify the presence of any risk factors for UTI in this population.

## Figures and Tables

**Figure 1 jcm-09-01531-f001:**
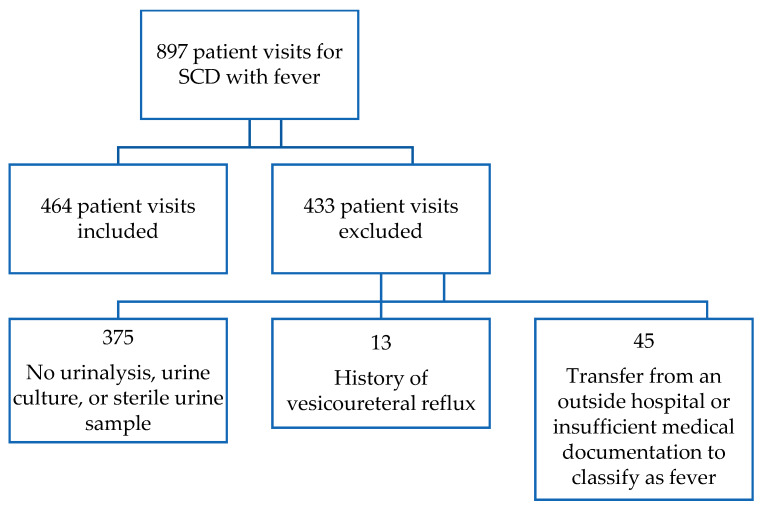
Study population. SCD: sickle cell disease.

**Table 1 jcm-09-01531-t001:** Demographics.

Characteristic	*n* = 167 (%)
Female	98 (58.7)
African American	159 (95.2)
**Type of Sickle Cell Disease**	
SS	109 (65.3)
SC	50 (29.9)
S-Beta Thalassemia	6 (3.6)
Other	2 (1.2)
**Variables**	**Median (IQR)**
Age (Months)	11 (7–20)
Maximum Temperature (°C)	39.3 (38.7–39.5)
White Blood Cell Count (k/mm^3^)	12.3 (8.9–18.1)
Hemoglobin (gm/dL)	9.5 (8.2–10.4)
Platelets (k/mm^3^)	310.5 (238–378)
Length of Stay (days)	2 (2–3)

**Table 2 jcm-09-01531-t002:** Characteristics of patients with and without urinary tract infection (UTI).

Characteristic	UTI *n* = 19	No UTI *n* = 445	*p*-Value
**Type of Sickle Cell Disease**	***n* (%)**	***n* (%)**	
SS	12 (63.2)	291 (65.4)	0.6
SC	6 (31.6)	139 (31.2)	
S-Beta Thalassemia	1 (5.3)	12 (2.7)	
**Variables**	**Median (IQR)**	**Median (IQR)**	
Age in Months	19 (12–30)	17 (10–28)	0.42
Maximum Temperature (°C)	39.5 (39.2–40.0)	39.3 (38.7–39.6)	0.13
White Blood Cell Count	13.8 (8.2–20.4)	13.6 (9.2–18.8)	0.82
**Urine Results Variables**	***n* (%)**	***n* (%)**	
White Blood Cell Count >5	12 (63.2)	38 (8.5)	<0.001
Leukocyte Positive	12 (63.2)	48 (10.7)	<0.001
Nitrite Positive	6 (31.6)	4 (0.9)	<0.001

**Table 3 jcm-09-01531-t003:** Urine culture results (*n* = 19 patient visits).

Patient Visit	Age in Months	Type of SCD	Name of First Urine Pathogen	Name of Second Urine Pathogen
1	11	S-beta thal	*Candida albicans*	*Enterococcus faecium*
2	31	SS	*Escherichia coli*	
3	14	SS	*Escherichia coli*	
4	19	SC	*Escherichia coli*	
5	26	SS	*Escherichia coli*	
6	30	SS	*Escherichia coli*	
7	11	SS	*Escherichia coli*	
8	16	SC	*Escherichia coli*	
9	22	SS	*Escherichia coli*	
10	17	SS	*Escherichia coli*	
11	38	SS	*Escherichia coli*	*Enterococcus faecalis*
12	45	SC	*Escherichia coli*	
13	15	SC	*Escherichia coli*	
14	11	SC	*Escherichia coli*	
15	12	SS	*Escherichia coli*	
16	25	SS	*Klebsiella pneumonia*	
17	3	SS	*Klebsiella pneumonia*	
18	47	SC	*Morganella moganii*	*Enterococcus faecalis*
19	22	SS	*Proteus mirabilis*	

**Table 4 jcm-09-01531-t004:** Generalized estimate model: factors associated with the presence of UTI.

Parameter	Estimates	Standard Error	95% Confidence Interval
Age in Months	1.01	0.03	0.95–1.07
Maximum Temperature (°C)	1.6	0.57	0.78–3.2
White Blood Cell Count	0.99	0.03	0.94–1.05

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
