# Peer review of "Urinary Tract Infection in Febrile Children with Sickle Cell Disease Who Present to the Emergency Room with Fever"

_jcm, 2020, doi:10.3390/jcm9051531_

Round 1

Reviewer 1 Report

In this article the authors proposed to measure the incidence of urinary tract infection (UTI) in febrile children with sickle cell disease (SCD) and under 4 years old in order of describe the risk factors for UTI in this population. They found an incidence equal of children without SCD.

The authors describe 19 children with fever and UTI and measure the incidence. It was equal to the “normal” children population.

But it would be interesting to know a little more about:

  • The renal status of these children compare to the reference population (plasmatic creatinine or glomerular filtration rate, albuminuria or proteinuria before and/or after UTI diagnosis).
  • The type of bacteria found in the urine,
  • The authors say UTI were increased in SCD patients because of abnormal urinary acidification and inability to concentrate the urine. What’s about these 19 children? Have these children, those biological markers?
  • What is the treatment (antibiotics?) and the outcome (renal function for example) of the 19 patients?
  • Author’s second aim was to describe the risk factors for UTI. Could they propose some risk factors despite they can not conclude in their study because of the smallest of the population?

The authors say, “the current practice is to perform a urine analysis in SCD patients with fever as part of the diagnosis”. But because of the difficulties to perform a clear catch sample of urine without invasive procedure, they want to identify risk factors. What will propose the authors for the children with or without these risk factors?

The authors report the prevalence of UTI in tropical countries and explain the role of environmental temperature but it would be interesting to compare the hygiene rules followed by the different populations and perhaps the difference of bacteria or parasite environment due of the tropical area.

Author Response

Response to reviewer #1:

In this article the authors proposed to measure the incidence of urinary tract infection (UTI) in febrile children with sickle cell disease (SCD) and under 4 years old in order of describe the risk factors for UTI in this population. They found an incidence equal of children without SCD.

The authors describe 19 children with fever and UTI and measure the incidence. It was equal to the “normal” children population.

But it would be interesting to know a little more about:

#1Comment: The renal status of these children compares to the reference population (plasmatic creatinine or glomerular filtration rate, albuminuria or proteinuria before and/or after UTI diagnosis).

Response: We thank the reviewer for this insightful comment. While it would certainly be   interesting to have this information, due to the retrospective nature and aim of the study this data was not be collected. However, I we did collect urinalysis results on each patient visits, and of the 19 with a UTI, 3 did have proteinuria.  

We have now added this comment to the results: Page 4, line 143-144.

There were 3 patients who had proteinuria seen in their urinalysis.

#2 Comment: The type of bacteria found in the urine.

Response: We thank the reviewer for this suggestion. We have now revised and added this information in the manuscript as an additional Table 3. Page 4, line 144-145, Page 6, line 158.

The most common bacteria found in the urine was Escherichia Coli

#2 Comment: The authors say UTI were increased in SCD patients because of abnormal urinary acidification and inability to concentrate the urine. What’s about these 19 children? Have these children, those biological markers?

Response: While previous studies have reported that patients with sickle cell anemia have an abnormal urinary acidification and inability to concentrate the urine causing them to have increased occurrence of UTI, (References 3-5,16), we did not study this specific question.

We have now added this comment to the discussion: Page 8, line 179-180.

While it would be interesting to see the effect of renal dysfunction on rates of UTI in children with sickle cell disease this was not within the scope of this ED based study.

#3 Comment: What is the treatment (antibiotics?) and the outcome (renal function for example) of the 19 patients?

Response: All patients were started on antibiotics. This information has now been added to the second paragraph of the results section as follows: Page 4, line 137-139.

All of the patients with a UTI were African American females with a median age of 19 months and they were all treated with antibiotics during the visit.

Given the retrospective nature and ED location of the study, we did not systematically follow outcomes especially since patients followed up with their pediatrician after resolution of their UTI. However, we are currently performing a follow-up study to examine the rate of asymptomatic bacteriuria and its relation to history of UTI in children with sickle cell anemia.

#4 Comment: Author’s second aim was to describe the risk factors for UTI. Could they propose some risk factors despite they can not conclude in their study because of the smallest of the population?

Response: Although in our study we were unable to find risk factors associated with an UTI in children with sickle cell anemia, we expect that the risk factors would be similar to that in non sickle cell patients with a fever namely age and gender (< 2 years in females and < 6 months in circumcised males and < 1 year in uncircumcised males).

#5 Comment: The authors say, “the current practice is to perform a urine analysis in SCD patients with fever as part of the diagnosis”. But because of the difficulties to perform a clear catch sample of urine without invasive procedure, they want to identify risk factors. What will propose the authors for the children with or without these risk factors?

Response: We thank the reviewer for this suggestion. We have now added this comment to the discussion: Page 8, Line 211-215.

If our study results are confirmed in a larger sample, then clinicians could potentially obtain catheterized urine cultures only in the at-risk population of children with SCD using the same criteria as for the children without sickle cell disease. Additionally, a screening urine analysis in triage with a bagged urine specimen in the at-risk children could guide the need for an invasive catheterized urine culture [Lavelle JM, Blackstone MM, Funari MK, et al. Two-step process for ED UTI screening in febrile young children: Reducing catheterization rates. Pediatrics. 2016;138(1)].

#6 Comment: The authors report the prevalence of UTI in tropical countries and explain the role of environmental temperature but it would be interesting to compare the hygiene rules followed by the different populations and perhaps the difference of bacteria or parasite environment due of the tropical area.

Response: We thank the reviewer for this insightful comment. We agree with the reviewer that impact of hygiene practices of different populations on UTI rates would be interesting question to explore especially given the different climates between tropical and temperate regions. Further cultural differences in hygiene practices have been reported to impact UTI rates in adults. (Das et al reported higher UTI rates in women using non disposable sanitary napkins). We have now added this comment to the discussion: Page 8, Line 197-200.

Although higher rates of UTI have been reported in adults in tropical countries due to differences in hygiene practice, the impact of cultural and population differences in hygiene practice on rates of UTI in children with sickle cell disease is unknown and needs to be explored further [Das P, Baker KK, Dutta A, et al. Menstrual hygiene practices, WASH access and the risk of urogenital infection in women from Odisha, India. PLoS ONE. 2015;10(6)].

Reviewer 2 Report

To the authors

In this paper the authors present a retrospective study reporting the number of children with SCD less than 4 years of age who presented at emergency department for fever (2012-2017) and who had urinary tract infection (UTI).

The aim of the study was to determine in a young SCD population the incidence of UTI and the risk factors. They report an incidence of 4.1% comparable to the incidence in non-SCD children reported in the literature and no authentified risk factor for UTI in this very young SCD population

There are many major concerns in this manuscript : the number of UTI reported is very low (n=19 within 5 years) and the choice to reserve the study to children younger than 4 years of age is not understandable. The definition of UTI was not clear « a UTI was therefore defined as a culture proven uropathogen of 10,000-100,000 CFU/ml or greater than 100,000 CFU/ml with 2 or fewer isolates ». The incidence calculation method is not described and the chapter on statistical analysis does not correspond to the tests carried out. Moreover, it is difficult to compare to non-SCD children because all young SCD-children must come to the emergency room in case of fever which is not the case for other children

Title

the term Sickle Cell Anemia is reserved to SS, Sb0 patients and SC, Sbthal. Thus, only the term SCD should be used here

Results

The calcul for incidence should be precised

Table 2 (urine results variables) : there is probably an inversion between UTI and non UTI and results presented only in 12 cases (but 19 UTI ?)

References

Ref 4 Journal name is missing

To the Editor

In this paper the authors present a retrospective study reporting the number of children with SCD less than 4 years of age who presented at emergency department for fever (2012-2017) and who had urinary tract infection (UTI).

The aim of the study was to determine in a young SCD population the incidence of UTI and the risk factors. They report an incidence of 4.1% comparable to the incidence in non-SCD children reported in the literature and no authentified risk factor for UTI in this very young SCD population

There are many major concerns in this manuscript : the number of UTI reported is very low (n=19 within 5 years) and the choice to reserve the study to children younger than 4 years of age is not understandable. The definition of UTI was not clear « a UTI was therefore defined as a culture proven uropathogen of 10,000-100,000 CFU/ml or greater than 100,000 CFU/ml with 2 or fewer isolates ». The incidence calculation method is not described. Moreover, it is difficult to compare to non-SCD children because all young SCD-children must come to the emergency room in case of fever which is not the case for other children

For these reaons, I consider this paper as not suitable for publication in JCM

Sincerely yours

Author Response

Responses to reviewer #2:

In this paper the authors present a retrospective study reporting the number of children with SCD less than 4 years of age who presented at emergency department for fever (2012-2017) and who had urinary tract infection (UTI).

The aim of the study was to determine in a young SCD population the incidence of UTI and the risk factors. They report an incidence of 4.1% comparable to the incidence in non-SCD children reported in the literature and no authentified risk factor for UTI in this very young SCD population

#1 Comment: There are many major concerns in this manuscript: the number of UTI reported is very low (n=19 within 5 years) and the choice to reserve the study to children younger than 4 years of age is not understandable.

Response: We thank the reviewer for this statement. We chose 4 years as a cut-off because children under this age frequently require catheterization due to lack of complete potty training. The procedure is associated with pain and discomfort in children. Further, these children and their families frequently have increased fear and anxiety regarding the catheterization especially since they may present to the ED with fever multiple times in a year thus requiring repeated catheterizations.  Hence, we felt it was important to study the rates and risks for a UTI in this age group so we can target only the identified high-risk group for catheterized urine samples.

#2 Comment: 19 visits with UTIs in 18 children in a population of approximately 150 children with sickle cell followed at our center in 6 years appears in line with the expected incidence.

The definition of UTI was not clear « a UTI was therefore defined as a culture proven uropathogen of 10,000-100,000 CFU/ml or greater than 100,000 CFU/ml with 2 or fewer isolates ».

Response: We thank the reviewer for this clarification. We have been defined a UTI per AAP criteria as follows:

The American Academy of Pediatrics (AAP) defines a UTI based upon method of collection: greater than 50,000 colony forming units per milliliter (CFU/ml) of an uropathogen for a sample obtained via catheterization or suprapubic aspiration with a positive UA (presence of pyuria and/or bacteria) [Roberts KB, Downs SM, Finnell SME, et al. Reaffirmation of aap clinical practice guideline: The diagnosis and management of the initial urinary tract infection in febrile infants and young children 2-24 months of age. Pediatrics. 2016; 138(6)]. Our laboratory reports urine CFU as less than 10,000, 10,000-100,000, or greater than 100,000 CFU/ml per pathogen.

We have now added this comment to the definition: Page 3, Line 95-97.

Due to this limitation, a UTI was therefore defined as a urine culture proven greater than or equal to 10,000 CFU/ml of at least one uropathogen.

#3 Comment: The incidence calculation method is not described and the chapter on statistical analysis does not correspond to the tests carried out.

Response: We thank the reviewer for this important comment. We recognize that since we included patient visits (rather than patients due to multiple visits during study period by each patient), our reported results are rates of UTI rather than incidence. We have now removed the word “incidence” from the title and from all of the manuscript and replaced it with “rates”. Further we have clarified all of the data analysis in the write up now as follows: Page 3, Line 114-116 and Page 4, Line 124-128.

The rate was measured by the frequency of which a UTI occurred in a defined population over 6 years. This was calculated with number of UTIs over the total number of visits.

Two group comparisons on UTI for non-normally distributed continuous variables such as age in months, maximum temperature °C and, white blood cell count (wbc) were compared using the Wilcoxon rank sum test. Generalized Estimating Equation for logistic regression was used to study the effect of different predictors (age of months, wbc, and maximum temperature) on the likelihood of UTI for the same subjects with repeated visits.

The Generalized estimating equation was performed and stated in the last sentence of the results section but now moved to a table 4. Page 4, line 145-146 and Page 7, line 160-161.

Based on the Generalized Estimating Equation for logistic regression analysis, age, maximum temperature and white blood cell count were not associated with a UTI (Table 4).

#4 Comment: Moreover, it is difficult to compare to non-SCD children because all young SCD-children must come to the emergency room in case of fever which is not the case for other children

Response: We thank the reviewer for this comment. We agree that while all SCD children present to the ED when they have a fever, this is not necessarily true of non SCD patients. This is one of the motivations for our study. Every child with SCD are being tested for UTI when they present with a fever when other non SCD wouldn’t have been if they did not meet the high-risk criteria. If the incidence of UTI is very low in children with SCD, it might suggest this testing may not be required in all patients with SCD and fever.

#5 Comment: Title: the term Sickle Cell Anemia is reserved to SS, Sb0 patients and SC, Sbthal. Thus, only the term SCD should be used here

Response: We thank the reviewer for this insightful comment. Accordingly, we have now altered the title to reflect this change and throughout the manuscript. Page 1, line 2-4.

Urinary Tract Infection in Febrile Children with Sickle Cell Disease who Present to the Emergency Room with Fever.

#6 Comment: Results: The calcul for incidence should be precise

Response: We thank the reviewer for this comment. We have now revised the manuscript to include the precise calculation for the rate/incidence as follows under data analysis: Page 3, line 114-116.

The rate was measured by the frequency of which a UTI occurred in a defined population over 6 years. This was calculated with number of UTIs over the total number of visits.

#7 Comment: Table 2 (urine results variables) : there is probably an inversion between UTI and non UTI and results presented only in 12 cases (but 19 UTI ?)

Response: It is correct that we had a total of 19 patient visits with a UTI. In table 2, we had only 12 UA which had greater than 5 WBC in the urine, the other 7 less than 5 WBC in the urine. Again, of the 19 UTI, 12 were positive for leukocyte esterase while 7 were negative. Only 6 had a positive nitrate while the other 13 were negative for nitrates.

#8 Comment: References: Ref 4 Journal name is missing

Response: We thank the reviewer for bringing this error to our attention and apologize for the same. We have now corrected this error in the references section as follows: Page 9, line 252-253.

Bruno D, Wigfall DR, Zimmerman SA, Rosoff PM, Wiener JS. Genitourinary Complications of Sickle Cell Disease. Journal of Urology. 2001; 166(3):803-811.

Round 2

Reviewer 1 Report

Thanks for the changes of the manuscript.

Reviewer 2 Report

The paper has been significantly improved. I suggest to add gender in the Table 2